# Demonstration of angular-momentum-resolved electron energy-loss spectroscopy

A. H. Tavabi [1,7], P. Rosi [2,7], G. Bertoni [2] ✉, E. Rotunno [2] ✉, L. Belsito [3], A. Roncaglia [3], S. Frabboni [2,4], G. C. Gazzadi [2], E. Karimi [5], P. Tiemeijer [6], R. E. Dunin-Borkowski [1] & V. Grillo [2]

Rotational invariance is a fundamental aspect of symmetry in scattering processes from atomic potentials. Here, we present an approach for measuring orbital angular momentum (OAM), a key descriptor of rotational symmetry, during measurements of atomic transitions. We use an electron optical OAM sorter in combination with electron energy-loss spectroscopy and model-based fitting to separately measure the π* and σ* antibonding transitions in hexagonal boron nitride on the atomic scale. This approach also offers prospects for efficient and atomically-resolved magnetic chiral dichroism measurements.

Electron energy-loss spectroscopy (EELS) in the transmission electron microscope (TEM) is one of the most powerful and versatile measurement tools in materials science[1], allowing the composition, valence state and occupation of electronic levels in materials to be investigated at the atomic scale. In such measurements, the electromagnetic nature of electron scattering allows energy-loss processes to be described in terms of optical absorption, with a single probe providing access to energies ranging from a few meV (*e.g.*, phonon excitations)[2] to keV (*e.g.*, core electron ionizations)[3,4].

In quantum electrodynamics, the absorption of a virtual photon produced by a fast incident electron is approximately equivalent to real photon absorption[5]. This analogy can be developed further by considering that, in the selection rules of optical absorption, the polarization of a real photon is substituted by a change in the momentum of the fast electron[6]. Based on this analogy, the selection rules for atomic transitions, expressed in terms of circular polarization of the photon, can be written in terms of the component of orbital angular momentum (OAM) of the electron $L_z$ in the propagation direction $z$.

The eigenvalues of this operator, $\hat{L}_z|\ell\rangle = \ell\hbar|\ell\rangle$, form a discrete spectrum characterized by the winding number $\ell$, whose eigenstates $|\ell\rangle$ are vortex beams[7–11]. Different methods have been proposed for measuring OAM[12–15], such as diffracting the electron beam using a pitchfork hologram[16]. However, the latter technique is relatively inefficient and does not fully separate the radial and angular components.

A better approach is provided by the so-called OAM sorter, an electron optical device that performs a log-polar conformal transformation to map OAM onto a linear dispersion that can be measured by means of diffraction in the TEM[17–19]. In the stationary phase approximation, this transformation can be achieved by using two electrostatic phase elements[20], labelled S1 and S2 in Fig. 1a, which first impart the transformation phase (S1) and then compensate for it (S2) after diffraction[21]. Such a measurement relies on perfect alignment of the elements. Despite the complexity of the setup, it has achieved a remarkable level of accuracy, as well as partial automation through neural-network-assisted alignment[22], resulting in an experimental resolution of $\Delta\ell \approx 1.1\,\hbar$ in vacuum[23].

The simultaneous analysis of electron energy-loss and OAM dispersion of inelastically scattered electrons, denoted OAM-EELS, has been predicted to enable innovative experiments in different scientific fields, encompassing plasmonic systems[24], biomolecular systems[25] and magnetic materials[26,27]. Here, we present the experimental realization of a combined OAM-EELS measurement by placing an OAM sorter in the post-specimen section of a TEM. The OAM spectrum is subsequently energy-dispersed, allowing for

[1]Ernst Ruska-Centre for Microscopy and Spectroscopy with Electrons, Forschungszentrum Jülich, 52425 Jülich, Germany. [2]Istituto Nanoscienze, Consiglio Nazionale delle Ricerche, Via G. Campi 213/A, 41125 Modena, Italy. [3]Istituto per lo Studio dei Materiali Nanostrutturati, Consiglio Nazionale delle Ricerche, Via P. Gobetti 101, 40129 Bologna, Italy. [4]Università di Modena e Reggio Emilia, Via G. Campi 213/A, 41125 Modena, Italy. [5]Department of Physics, University of Ottawa, Ottawa, ON K1N 6N5, Canada. [6]Thermo Fisher Scientific, PO Box 80066, 5600 KA Eindhoven, the Netherlands. [7]These authors contributed equally: A. H. Tavabi, P. Rosi. ✉e-mail: giovanni.bertoni@cnr.it; enzo.rotunno@cnr.it

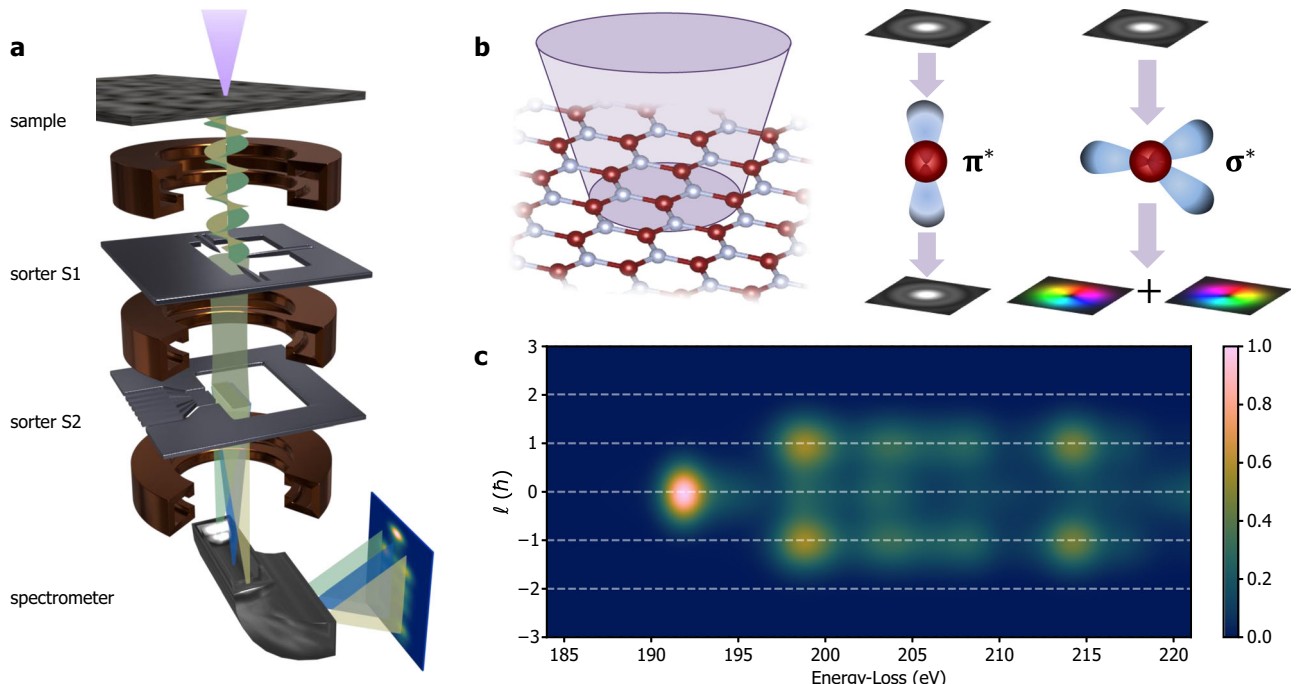

**Fig. 1 | Experimental setup for OAM-EELS. a** Schematic diagram of the experimental setup, which includes two sorter elements[19] (S1 and S2) after the sample plane and a spectrometer to realize OAM-EELS double dispersion. **b** Schematic diagram showing electron beam illumination of h-BN with a finite convergence semi-angle and the effect on the electron wavefunction of single scattering at the B K-edge. **c** Simulated OAM-EELS measurement for an OAM resolution of $\Delta\ell = 1.1\hbar$.

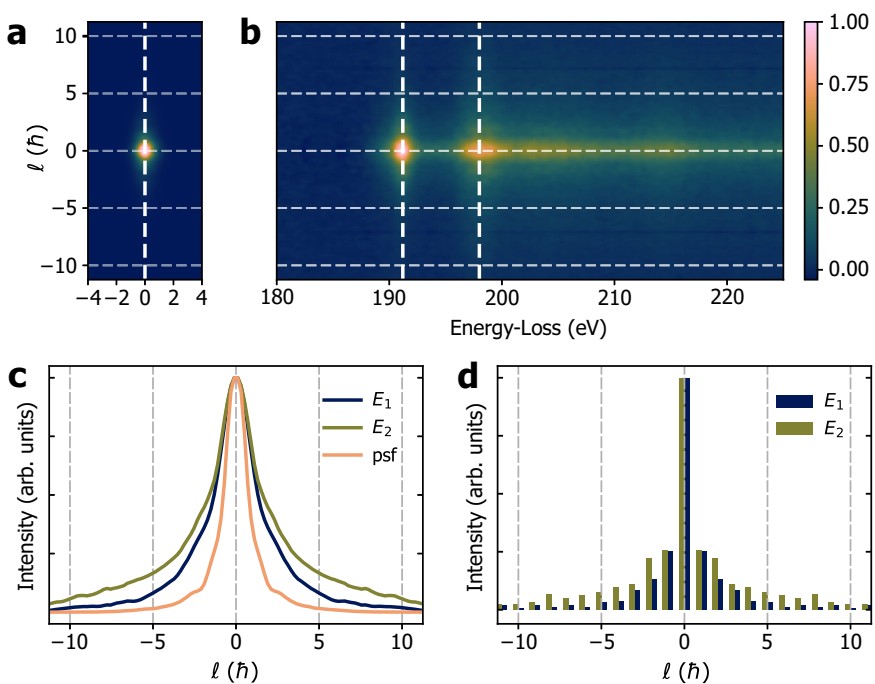

**Fig. 2 | Experimental OAM-EELS from the B K-edge. a** Experimental OAM-EELS zero-loss spectrum and **b** B K-edge spectrum after background removal. **c** Corresponding experimental OAM profiles at $E_1 = 191$ eV (dark blue line) and $E_2 = 198$ eV (olive green line) from the B K-edge compared with the ZL profile (psf) at 0 eV (orange line). **d** Discretized OAM profiles at $E_1$ (dark blue bars) and $E_2$ (olive green bars) after psf deconvolution.

the simultaneous recording of OAM and electron energy-loss in a single measurement (Fig. 1a). We demonstrate the approach by studying the B K-edge in hexagonal boron nitride[28] (h-BN) (Fig. 1b). This van-der-Waals-layered material contains bonds that are based on $sp^2$ hybridization of its atomic orbitals, resulting in σ and π

bonding orbitals and σ* and π* antibonding orbitals. Transitions from the occupied 1s state ($m = 0$) to the unoccupied hybridized orbitals are characterized by magnetic quantum numbers $\Delta m = +1, -1$ (σ*) and $\Delta m = 0$ (π*)[29] when the quantization axis is orthogonal to the basal plane (*i.e.*, parallel to the *c* axis). The

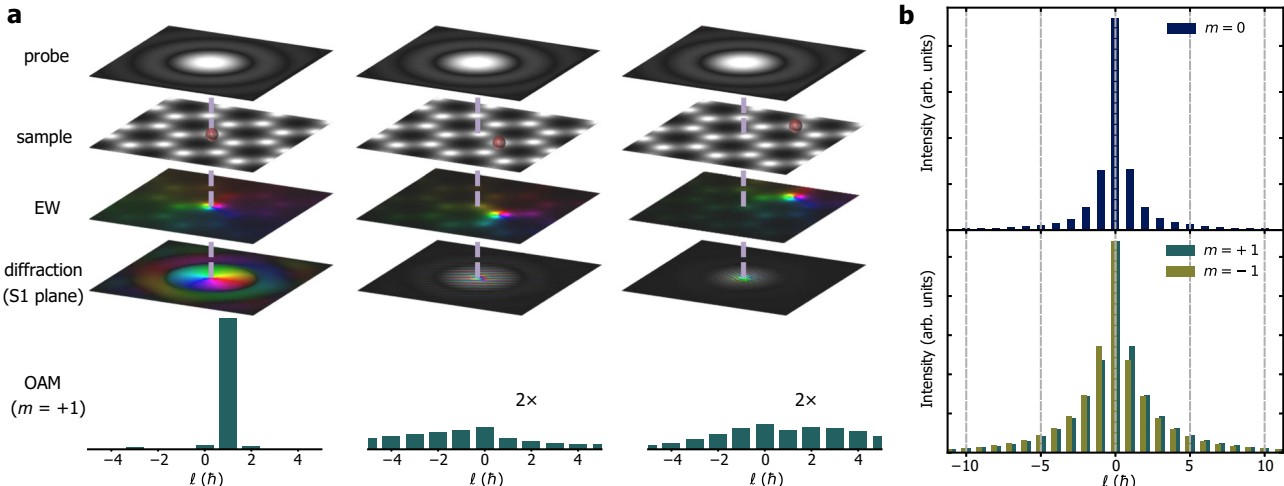

**Fig. 3 | Numerical model for simulating scattering delocalization. a** Effect of probe size and scattering delocalization for $m = +1$ for a B atom on the optical axis of the OAM sorter (left column) and for two B atoms off-axis. Each other B atom in the illuminated area (shown) emits its own inelastically-scattered wave. In the far field, the phase of the off-axis atoms is distorted. The OAM sorter records an incoherent sum $\Gamma_m(\ell)$ of their contributions (teal bars, the off-axis atom intensities have been multiplied by two). **b** Simulated OAM profiles $\Gamma_m(\ell)$ for $m = 0$ (dark blue bars) and $m = \pm1$ ($m = +1$ teal bars, $m = -1$ olive green bars).

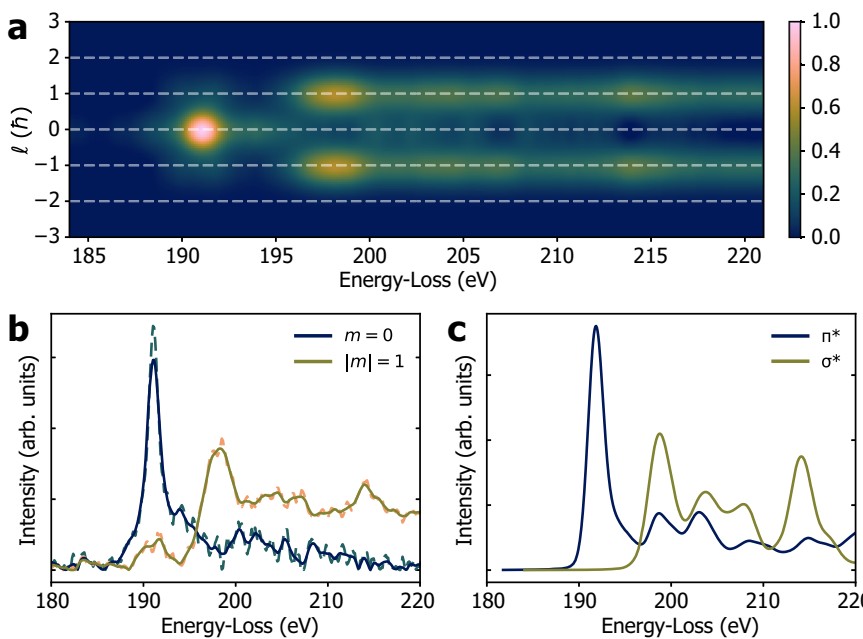

**Fig. 4 | Results of model-based fitting. a** Reconstructed OAM-EELS $I(E,m)$ spectrum after deconvolution. A Gaussian modifier with $\Delta\ell \approx 1.1\hbar$ was used for comparison with Fig. 1c. **b** EEL spectra $c_m(E)$ obtained from the experiment. The dashed lines are results of the fit ($m = 0$ teal dashed line, $m = +1$ orange dashed line). The full lines are smoothed versions ($m = 0$ dark blue line, $m = +1$ olive green line). **c** Corresponding EELS/XANES spectra for $m = 0$ (π*, dark blue line) and $|m| = 1$ (σ*, olive green line) obtained from ab initio DFT calculations.

transitions occur at two different energies, which differ by ~6 eV for the B K-edge. In a standard EELS experiment, in which only energy-loss is recorded, the edge onset is well defined, but the post-edge features are superimposed[29]. In an OAM-EELS experiment with the electron beam aligned with both the optical axis of the OAM sorter and the quantization axis, these transitions correspond to scattered electron winding numbers of $\ell = \pm1$ and $\ell = 0$, respectively (Fig. 1c). For simplicity below, $m$ is used to refer to the intrinsic angular momentum along $z$ of the scattered electron, while $\ell$ is used to refer to the OAM values measured with the OAM sorter.

## Results

By leveraging the OAM dispersion, we aim to separate the spectral components π* ($m = 0$) and σ* ($m = \pm1$) in the post-edge region. Figure 2a shows a zero-loss (ZL) OAM-EELS spectrum recorded in vacuum. Figure 2b shows a raw OAM-EELS experimental measurement of the B K-edge after background subtraction. The two raw OAM profiles at $E_1 = 191$ eV and $E_2 = 198$ eV, which lie slightly above the onsets of the π* and σ* components, respectively, are presented in Fig. 2c. The $E_2$ OAM profile is broadened with respect to the $E_1$ OAM profile due to the $m = +1$ and -1 contributions, while the $E_1$ OAM profile is sharper due to the presence of only the $m = 0$ contribution. The OAM profile of the ZL

peak is also shown in Fig. 2c. As it was acquired in vacuum, and since chromatic effects of the sorter can be neglected in this energy range (see Supplementary Information), it can be regarded as the point spread function (psf) of the OAM sorter, which contributes to the OAM resolution measured by the device. The OAM spectrum is relatively broad and extends beyond the expected $\ell = 0$ and $\ell = \pm 1$ contributions. Figure 2d shows the resulting quantized OAM profiles $I(E_1, \ell)$ and $I(E_2, \ell)$ after deconvolution by the experimental psf using multiple linear least-squares fitting[30,31]. After deconvolution, the OAM profiles still have extended tails, with a maximum at $\ell = 0$.

We show here that this broadening is associated primarily with delocalization of the inelastic scattering. We consider a simple model of a single inelastic scattering event. This model accounts for the probe size, localization of the inelastic scattering and uncertainty in the probe position. It has a single fitting parameter, *i.e.*, the probe defocus, which can be adjusted against the $E_1$ OAM profile, which is expected to have only the $m = 0$ contribution.

After scattering, the fast electrons are in a non-separable state $|\psi_{a,m}\rangle$ with the atomic excitations[32]. In the dipole approximation, this state takes the approximate form:

$$\left|\psi_{a,m}\right\rangle = f_{a,m}(\mathbf{r}) \cdot \left|\psi_p\right\rangle, \tag{1}$$

in real space $\mathbf{r} = (x, y)$, where $|\psi_p\rangle$ is the electron wavefunction calculated (typically using a multislice or Bloch wave algorithm[33]) at the depth of the scattering atom and $f_{a,m}(\mathbf{r})$ is the scattering function of atom $a$[34–36]. Interaction of the probe with each atom produces a separate set of waves $|\psi_{a,m}\rangle$ with varying $m$. In the present experiment, the transitions are limited to $1s \rightarrow \pi^*$ ($\Delta m = 0$) and $1s \rightarrow \sigma^*$ ($\Delta m = \pm 1$). The OAM spectrum that is produced by the OAM sorter is an incoherent sum $\Gamma_m(\ell) = \sum_a |\langle \ell | \psi_{a,m}\rangle|^2$ of contributions from all excited atoms $a$. It can be obtained numerically by using a statistical Monte Carlo method, which is described in the Supplementary Information.

Figure 3a shows numerical calculations of the contributions to an OAM spectrum for the $m = +1$ transition from three different atoms, one of which is located on the optical axis and two off-axis. Only atoms on the optical axis produce a narrow decomposition centered on the $\ell = +1$ value in the OAM sorter. The off-axis atoms contribute to a broad spectrum with a dominant $\ell = 0$ contribution. Figure 3b shows the resulting $\ell = 0$ and $\ell = \pm 1$ OAM spectra after scattering through the sample, taking into account the thickness of the sample ($t \sim 8$ nm, measured from the experimental low-loss spectrum) and the dimensions of the probe (with a 5.4 mrad convergence semi-angle). This sample thickness ensures a sufficient signal in the B K-edge, while keeping multiple scattering negligibly low[26]. The profiles are in good agreement with the corresponding experimental profiles shown in Fig. 2c, d. The simulated OAM profiles $\Gamma_m(\ell)$ shown in Fig. 3b were finally fitted to the experimental data $I(E, \ell)$, in order to obtain a set of coefficients $c_m(E)$ corresponding to EEL spectra at different $m$, according to the equation:

$$I(E, \ell) = \sum_m c_m(E)\Gamma_m(\ell). \tag{2}$$

Figure 4a shows the resulting reconstructed OAM-EEL spectrum $I(E, m) = \sum_m c_m(E)$ for comparison with Fig. 1c. Figure 4b shows the $m = 0$ and $m = \pm 1$ EEL spectra. For comparison, Fig. 4c shows $m = 0$ and $m = \pm 1$ EELS/XANES spectra obtained from ab initio DFT calculations using a core-hole approximation within Quantum ESPRESSO[37,38].

The overall agreement between experimental and theoretical spectra demonstrates the quantitative nature of the method, with key spectral features matching both in relative intensity and in shape. A small residual peak at 190 eV in the $m = \pm 1$ ($\sigma^*$) spectrum is observed, but its influence is minimal, as confirmed by quantitative metrics such as peak ratio analysis and normalized cross-correlation, as detailed in

the Supplementary Information. We estimate a cross-talk contribution of approximately 11%, which remains within acceptable limits for most applications, including magnetic dichroism measurements. Possible sources of this discrepancy include residual channelling effects, minor misalignments affecting the sorter psf, or limitations in the physical model. While our simplified model is sufficient for the present study, future refinements could incorporate a full inelastic multislice approach for enhanced accuracy.

## Discussion

Our experimental results reveal an OAM spectrum that is characterized by a broad distribution of $\ell$, rather than by the anticipated discrete contributions at $\ell = 0$ and $\ell = \pm 1$, due to the inelastic delocalization. At energies in the range of a few hundred eV, the inelastic scattering function has long-range tails that extend beyond the nearest-neighbour atomic distances. This means that atoms away from the immediate probe position contribute to the scattering signal, reducing the localization of the interaction. Secondarily, technological constraints inherent to the current OAM sorter also impact on the achievable resolution. Specifically, aberrations introduced by imperfections of its elements distort the OAM spectrum. These aberrations are mostly pronounced at larger scattering angles, at which the performance of the OAM sorter degrades. To minimize their influence, the convergence semi-angle of the STEM probe was restricted to a modest value of 5.4 mrad, by means of a small diaphragm. Whereas this adjustment reduces the effects of aberrations, it also increases the probe size, leading to contributions from atoms located away from the optical axis. It also reduces beam current, necessitating longer acquisition times that, in turn, lead to sample drift.

Since the definition of angular momentum depends strongly on the choice of the pole, which corresponds here to the optical axis of the OAM sorter, atoms that are positioned away from the optical axis produce a broad OAM spectrum despite having a well-defined value of $m$[39–41]. For the same reason, a small displacement of the probe from the optical axis broadens the OAM spectrum. An estimate of the magnitude of this effect can be obtained from the Roberston uncertainty relationship $\sigma_A \sigma_B \geq \frac{1}{2}\left|\left\langle\left[\hat{A}, \hat{B}\right]\right\rangle\right|$ between operators $\hat{A}$ and $\hat{B}$. The operator $\hat{L}_z$ and the position operator $\hat{x}$ in the orthogonal plane are related by the expression $[L_z, x] = i\hbar y$[42], while the uncertainties are related by $\sigma_\ell \sigma_x \geq \frac{\hbar}{2}\langle y\rangle$. In this expression, the uncertainties are expressed as standard deviations. $\sigma_x$ is the circularly symmetric width of $|\psi_{a,m}\rangle$, which can be calculated from the variance of the scattering intensity. In other words, for a scattering atom located at position $\langle y\rangle$ from the center of the probe and optical axis, the OAM spectrum is broadened by $\sigma_\ell \geq \frac{\hbar}{2}\frac{\langle y\rangle}{\sigma_x}$. An atom that is located exactly on the optical axis does not produce OAM broadening, i.e., $\langle y\rangle = 0$. As a result of the delocalization of inelastic scattering, off-axis atoms contribute to the OAM spectrum even if they are not directly under the probe tails. This effect can be described approximately by the expression $\langle y\rangle = \sqrt{\sigma_p^2 + \sigma_x^2}$, where $\sigma_p$ is the standard deviation of the probe intensity. The uncertainty in OAM dispersion can be written in the form:

$$\sigma_\ell \geq \frac{\hbar}{2}\sqrt{1 + \sigma_p^2/\sigma_x^2}. \tag{3}$$

In our experiment, the probe extends over $\sigma_p \sim 0.25$ nm at $\alpha = 5.4$ mrad. By considering an isolated B atom scatterer, $\sigma_x \sim 0.32$ nm, very close to the delocalization value $0.5\lambda\theta_E^{3/4}$ estimated by Egerton[43]. They contribute overall to $\sigma_\ell \geq 0.7\hbar$ for an isolated B atom. The evolution of the wavefunction within the crystal (channelling) contributes to the localization of the exit wave function on the atomic columns, reducing $\sigma_x$ to $\sim 0.05$ nm, as calculated from the standard deviation of the exit wave intensity obtained from multislice calculation, while increasing the overall indeterminacy of OAM to $\sigma_\ell \geq 2.55\hbar$. This value is in good

agreement with the experimental findings, for which $\sigma_\ell = 2.7\hbar$ based on the $E_1$ profile shown in Fig. 2d. The formula in Eq. 2 can be generalized to atomic number $Z$ in the 1s Bloch wave approximation[44], where the focusing of the propagating beam due to the atomic columns can be expressed semi-empirically as $\sigma_x \propto \sqrt{\frac{Z}{d_z^{5/4}}}$, with $d_z$ the separation of the atom in the column. Heavier atoms are expected to produce more peaked OAM profiles.

The primary limitation of the current OAM-EELS experiment stems from the fabrication precision of the first sorting element. However, these are not fundamental limitations but rather technological challenges that can be overcome by improving the OAM sorter design so that it works with larger convergence semi-angles, thereby reducing the size of the probe and the value of $\sigma_p$. Similarly, limiting the collection angle by using an aperture built into the device, which is equivalent to increasing the uncertainty $\sigma_x$, also has the effect of reducing the OAM broadening $\sigma_\ell$. While the intrinsic delocalization of inelastic scattering will always require model-based fitting for rigorous quantitative interpretation (see Supplementary Information), a more advanced experimental setup will reduce artifacts and simplify this process, further strengthening the method's reliability and applicability.

Achieving spatially resolved OAM-EELS measurements presents additional challenges compared to conventional EELS, as scanning the beam can disrupt the precise alignment of the TEM column and the OAM sorter. In particular, accurate alignment of the S1 and S2 phase plates is crucial to maintain optimal performance, especially when scanning is required, such as during spectrum image acquisition. Despite these challenges, OAM-EELS spectral images can be obtained by using the descan coils of the TEM, as outlined in the Methods section and demonstrated in the Supplementary Information (Figures S1 and S2). This second experiment was analysed using multivariate statistical analysis[45], in order to verify the presence of the same OAM components without using a model. In this low magnification example, we observed an effect of the sample edge. Interpreting in a quantitative way these observations require detailed structural models of the sample folding, and it is well beyond the scope of the present paper. However, the ability to measure a spatially resolved signal hints at future applications of atomic-resolution scanning experiments.

It is useful to compare OAM-EELS with other state-of-the-art techniques. The use of STEM-EELS, in combination with a corrected probe and a high-resolution spectrometer, permits symmetry-related features of excited states to be extracted by tracing the scattered intensity in a given energy-loss window using so-called orbital mapping[46–49]. However, only peaks that are well separated in energy in the EEL spectrum can be mapped and related to different orbitals, defects or dopants[50]. Furthermore, in order to distinguish and map different out-of-plane contributions, a side view of the sample is also needed, which is challenging to realize experimentally. In contrast, OAM-EELS allows spectral features to be separated directly at each probe position. A combination of this method with beam scanning leads to the prospect of mapping orbitals in the basal plane[42,43], allowing discrimination of the details of chemical bonds separately. From the perspective of atomic-resolution scanning experiments, it allows for post-selection of dipolar transitions even when using a large integration angle[51]. The ability to distinguish $\ell = \pm 1$ transitions is even more appealing for applications that involve dichroism. In an electron-photon analogy, electron magnetic circular dichroism (EMCD) is a counterpart of XMCD, a well-established X-ray absorption technique for measuring magnetic moments. OAM-EELS can be used to provide a spectrum of asymmetries between transitions with angular momentum exchange +1ℏ and -1ℏ, as in the $L_{2,3}$ ionization edges of magnetic $3d$ elements[26,52], to assess the spin population in a material[53]. Unlike

orbital mapping using STEM-EELS, in which spectra are $q$-integrated over the high convergence angle of the probe and the spectrometer collection angle, a different approach involves performing linear momentum dispersion experiments (qEELS)[54–56], which are in principle able to retrieve information about bond orientation. However, the very high momentum resolution that is then needed requires the use of nearly parallel illumination and a small pupil aperture (with a convergence semi-angle $\alpha \sim 0.1$ mrad), which is not compatible with an atomic-size probe. In comparison, the different commutation rules that are applied to OAM-EELS possess the advantage of lower uncertainty in OAM dispersion $\sigma_\ell$, while increasing the spatial resolution by reducing the probe dimension $\sigma_p$. Moreover, a residual signal from σ* at $q = 0$ is expected in $q$-resolved experiments due to the finite size of the aperture.

In summary, we have experimentally demonstrated angular-momentum-resolved electron energy-loss spectroscopy at the B K ionization edge in a thin h-BN sample. This technique allows direct measurement of a final state's orbital angular momentum and is made possible by combining EELS with the use of an OAM sorter and model-based fitting. As a result of the different commutation rules with respect to position, an OAM-EELS measurement is compatible with the use of an atomic-scale probe, in contrast to linear-momentum-resolved EELS. Beyond the separation of σ* and π* orbitals, the technique offers important prospects for the characterization of multipolar transitions and efficient EMCD measurements.

## Methods
### Experimental setup
In a first experiment, the B K-edge was recorded using an FEI Titan TEM at 300 kV ($C_s = 2.7$ mm), a Gatan Quantum spectrometer and an OAM sorter consisting of a first device (sorter unwrapper, S1) inserted in the objective aperture plane and a second device (sorter corrector, S2) inserted in the selective area plane (Fig. 1a). The microscope was operated in OAM-EELS mode, with energy dispersion on the horizontal ($x$) axis and OAM dispersion on the vertical ($y$) axis. In order to reduce carbon contamination, the sample was heated to 120 °C overnight in vacuum before insertion into the TEM. A convergence semi-angle $\alpha = 5.4$ mrad was used to reduce the effect of aberrations of the OAM sorter at high angles (probe size $\sigma_p \cong 0.25$ nm). The B K-edge was recorded at 1x binning with a 30 s total exposure time to reduce noise. A thin well-oriented [001] h-BN crystal was used. The energy dispersion was 0.03 eV per pixel on the $x$ axis and 0.1 ℏ per pixel on the $y$ axis. The latter value was measured by using a petal beam with an $\ell = -4, +4$ vortex generator in a separate study. A second experiment (see Figures S1 and S2) was conducted at 300 kV in a Thermo Fisher Spectra 300 TEM equipped with a cold field emission gun and a Selectris energy filter. The probe aberration corrector was switched on ($\sigma_p \cong 0.67$ nm at $\alpha = 1.8$ mrad), the image aberration corrector was switched off and the 'CoolTEM' software plugin was used to control the projector lens settings to enable free rotation of the diffraction pattern with respect to the spectrometer entrance aperture. The energy dispersion was set to 0.1 eV per pixel on the $x$ axis and 0.063 ℏ per pixel on the $y$ axis. In scanning mode, the descan coils were tuned to maintain the OAM sorter alignment for all scanning points. Spectra were recorded using a dwell time of 1 s per STEM pixel and a probe current of 600 pA.

### Data analysis
Each two-dimensional (2D) spectrum was first pre-processed, according the following procedure: 1) 2D rotation by 0.2° to orient the spectrum along $x$ and $y$; 2) Cropping of the 2D spectrum to 1840 × 256 pixels centered on the $\ell = 0$ line (or the center line in the 2D spectrum); 3) Background subtraction according to a power law function in the B

K-pre-edge region for every line in orbital momentum; 4) Suppression of X-ray spikes by using a threshold in intensity of 5 times the standard deviation of the data; 5) 4× binning in energy and 2× binning in OAM, followed by OAM symmetrisation with respect to $\ell = 0$, in order to obtain a 2D spectrum of 460 × 128 pixels with reduced noise. After pre-processing, custom Python routines were used to perform OAM profile extraction and deconvolution using multiple least-squares fitting, by making use of numpy and scipy libraries. The decomposition presented in the Supplementary Information was obtained with non-negative matrix factorization (NMF) with the 'sklearn' library. Theoretical B K-edge spectra were calculated using the 'xspectra' routine in Quantum ESPRESSO[37,38], after convergence of a ground state self-consisting cycle on a 3 × 3 superstructure of h-BN. A core-hole approximation was used with 0.5 holes in the 1 s core level to consider attraction of the hole, which reweights the spectral intensity towards lower energies, as seen experimentally.

## Model-based fitting

In order to analyse the experimental data, model-based fitting was used. This approach relies on a kinematic model of the exit wave at the sample plane, where the exit wave is expressed as a simple product of 2D functions: the 1 s Bloch state, the probe, and the atomic scattering factor, as detailed in the Supplementary Information. The exit wave is calculated for multiple configurations of atom positions, considering variations introduced by the stochastic nature of inelastic scattering events, indetermination of the probe position and sample drift. We employed a Monte Carlo approach to sample random configurations of illuminated atoms within a disk of radius 0.7 nm centred on the probe. The resulting angular momentum components were obtained by projecting the wave onto eigenstates of the OAM operator. These spectra were fitted to the experimental data to extract relevant parameters, most notably the defocus of the STEM probe. All functions were sampled on a square mesh grid of 512 × 512 pixels with a lateral dimension of 7.85 nm, corresponding to a resolution in the Fourier plane of 0.25 mrad. The STEM probe was calculated assuming $C_s = 2.7$ mm, in agreement with the nominal value for the microscope used for the experiment. The defocus was optimized within a 100 nm range around the Scherzer condition for a 300 kV beam. 1000 different configurations were considered for every calculation. All the calculations were performed using STEM_CELL software[57]. Source Data and code generated in this study are available[58].

## Data availability

Source Data file has been deposited in Zenodo under accession code https://doi.org/10.5281/zenodo.15470500.

## Code availability

The code generated in this study has been deposited in Zenodo under accession code https://doi.org/10.5281/zenodo.15470500.

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

## Acknowledgements

This work was supported by the European Union's Horizon programme through the project IMPRESS (grant agreement No. 101094299, A.H.T., G.B., E.R., L.B., A.R., R.E.D-B., V.G.) and the European Union's Horizon 2020 research and innovation programme through the project ESTEEM3 (grant agreement No. 823717, A.H.T., E.R., R.E.D-B.) and the project Q-SORT (grant agreement No. 766970, A.H.T., P.R., G.B., E.R, L.B., A.R., S.F., G.C.G., E.K., R.E.D-B., V.G.) and the project SMART-electron (grant agreement No. 964591, P.R., G.B., E.R., V.G.). We thank the Italian Ministry of University and Research Decree 128 – 21/06/2022, Infrastructure for Energy Transition and Circular Economy - iEN-TRANCE@ENL (contract No. IR0000027, G.B., E.R., L.B., A.R., V.G.). We are grateful to R.F. Egerton for fruitful discussions.

## Author contributions

V.G., A.H.T. and G.B. conceived and designed the experiment; E.K. conceived the idea of the OAM sorter; P.R., L.B., A.R. and G.C.G. created the phase structures; A.H.T., P.T., E.R. and S.F. contributed to the experimental setup; A.H.T. and P.T. collected the experimental data; G.B., E.R., P.T. and V.G. performed data analysis; G.B., E.R., V.G., A.H.T., P.R., and R.E.D.-B. wrote the paper.

## Competing interests

The authors declare no competing interests.
