## [Transparent Peer Review file · Nature Communications]

Demonstration of angular-momentum-resolved electron energy-loss spectroscopy

Corresponding Author: Dr Giovanni Berton

Version 0:

Reviewer comments:

Reviewer #1

(Remarks to the Author)

The paper presented experimental results on the orbital angular momentum resolved electron energy loss spectrum and their theoretical analysis. Despite the less than ideal instrumental resolution on the OAM state, the raw experimental data does show signs that the pre-edge spectral feature is associated with the dipole electronic transition involving exchange of zero unit of angular momentum and some of the post-edge spectral weight is associated with exchange of one unit of angular momentum. The results of multivariate statistical analysis merely confirm what one already can deduce qualitatively from the raw data, but adds additional confusion because it is well-known that the physical nature of different components are unclear in such statistical analysis. The discussion of the contribution to the OAM-resolved spectral signals covers possible OAM broadening through crystal diffraction and contribution from off-axis atoms, although I have some questions about the underlying assumptions, interpretation and implications. Overall, it is an interesting paper worthy publication after some essential modification. Below are my detailed comments:

- 1) Line 31: 'In the second quantization framework, the absorption of a virtual photon produced by a fast incident electron is approximately equivalent to real photon absorption' This reference to the second quantization framework is obscure, can the author provide a reference or more context to clarify the statement.
- 2) The raw experimental results are not reproduced in the main text, but Fig. 2 instead shows the spectral components deduced from non-negative matrix factorization algorithm processing (NMF). Given that NMF is a variation of the multivariable statistical analysis and I assume that it also suffers the usual problem of interpretability of the origins of the different decomposed components, it is not appropriate to show them as the main result in Fig. 2, instead of the raw data (shown in Fig. S1). As an example, can the authors explain the OAM dependence of the residual, which also display the same dependence as the so-called π^* components. In fact, what is the basis of identifying two similar looking components differently (the display of the scree plot in the supplementary materials could be useful here)? Incidentally, are the spectral intensities presented in Fig. 2a and 2c normalized in any way? If not, it is disturbing that the 'residual' has such a large intensity, indicating the significant failure of the OAM separation there.
- 3) Instead of NMF analysis in the main text (which is perfect for inclusion in SI), it maybe more profitable to present to present the raw data and a 'sharpened' version by deconvoluting out the instrumental OAM 'response function', given one is already collected at the zero energy loss (Fig S6b) and at pre-edge energy (S1b), or one can be recorded separately using the phase plates for different OAM channel. Alternatively, the raw OAM-EELS data can be compared directly with the properly convolved OAM-EELS spectrum from first principle calculation.
- 4) In lines 136-141, the authors discussed the extraction of bond orientation experiment qEELS experiments based on high momentum resolution and stated correctly that the use of nearly parallel illumination is incompatible with the use of atomic sized probes. In this context, the authors should also mention and comment on the convergent beam variation of the qEELS work, for example, that of ref 40 (an edge-on sample required), Hu et al. (2008) Ultramicroscopy, Vol 108, p405 (an edge-on sample not essential) and Xu et al (2023) Phys. Rev. Lett. 131, 186202 (an edge-on sample not required). It provides a complementary alternative to spatially resolved OAM-EELS analysis. This also applies to the relevant comments in lines 142-144 and 160-162.
- 5) In Figure 3, the consequence of inclusion of the inelastic scattered waves from the off-axis atoms is displayed

qualitatively. Do you mean schematically ? Given it is the single most important effect in canceling out the OAM-specific effect in a lot of EELS experiments tried so far, can we have a specific example, for example, from your Monte Carlo simulation mentioned in lines 98-110.

6) Assume that the off-axis contribution is included in the simulated OAM spectra Fig. 2d, can we also include the theoretical OAM spectra from the on-axis atom alone for comparison? I think that the data can be readily extracted from Fig. 1c.

7) A comparison of Fig. 1c and Fig. 2d seems to suggest that the observed broadening of the OAM distribution is largely due to the unavoidable contribution of the off-axis contribution of the neighboring atoms. Given that, how can the unknown non-zero OAM spectral modes be deduced correctly? (This is the prerequisite to realize the potentials stated in lines 147-156).

8) In Line 163, the prospect of multipolar transition is mentioned. Is there evidence for such transitions in the current B K-edge data ? How can such a transition be distinguished from dipolar contribution from off-axis atoms?

9) The phrase 'diffraction' has been used in two different contexts: one of them is used properly as in Lines 111-112, '... no diffraction effect.', i.e. the diffraction by the rest of the crystal post inelastic core loss scattering. On the other hand, the term 'diffraction' has been used to indicate the Fourier transform of the exit wave by the objective lens (line 333). The second usage is confusing, hence please amend.

10) Given the discussion, can the authors summarize at the end of the paper the essential experimental conditions for the observation of intrinsic OAM-resolved EELS? For example, what is the maximum thickness before the crystal diffraction needs to be taken into account? Can the OAM-EELS experiment have a simple interpretation if the symmetry axis of the sample does not align with the beam axis ?

Reviewer #2

(Remarks to the Author)

In this manuscript, the authors report an experimental demonstration of an orbital-angular-momentum sorter for EELS spectra in the TEM. The authors chose to acquire data on a highly anisotropic material, hexagonal BN, which features π^* and σ^* contributions energies separated by several eV. The non-negative matrix factorization statistical analysis approach was chosen to exploit the experimental data. The manuscript is well written, contains a short results section followed by a longer discussion about the interpretation of the results, the current limitations and perspectives. Although this demonstration is a technical tour de force, which validates the predictions from simulations, I have several comments below.

- This is a rather compelling experimental demonstration of angular-momentum-resolved EELS on a ~8 nm thick h-BN sample. This manuscript provides ground to the OAM-EELS approach, which has been introduced on a theoretical basis by the authors in previous publications and conference abstracts. The demonstration that the OAM sorter is working in a TEM has already been reported by the same group of authors (reference 18, Amir H. Tavabi et al., physical review letters 126, 094802 2021), and since the OAM-EELS theory is also available in literature, the highlight of this manuscript is really on the experimental demonstration of an OAM-EELS on h-BN.

The added-value of the experiment is evident to validate the simulations. Nevertheless, beyond this experimental demonstration, the manuscript does not provide much support of the expected impact that the OAM approach could bring when probing localized states, decorrelating and mapping hybrid orbitals, as mentioned in the main text. This is rather disappointing since these ideas are introduced at the beginning (Figure 1), but remain to be shown experimentally.

One of the major downside of this experimental demonstration is the lack of spatial resolution, which is one of the main advantages of using a TEM. As such, the sub-nanometer or atomic-scale resolution available in current TEM instruments is the missing ingredient for a truly impactful demonstration. The discussion contains the current limitations of the OAM-EELS approach and provides perspectives for this work. This is interesting, but at the same time clearly suggests that experimental difficulties remain to unleash the potential of this elegant approach, which is at the expense of the impact of the results. Beam scanning, for instance, is mentioned but not demonstrated because of physical or technical limitations of the sorter.

The lack of clear broad impact perspective that one could expect for a journal of this caliber is detrimental. Whilst the separation of OAM components is of interest, the broad impact of the OAM sorter for TEM analysis is not apparent in this manuscript. As a result, and again despite the scientific soundness of the work, this manuscript would be a much better fit to the audience of a more specialized journal.

- Overall, the results in Figure 2 should be explained and discussed further in the manuscript, as detailed in the following:

The use of the word "closely" in the sentence "The NMF-extracted experimental components closely mirror those predicted by DFT calculations..." is qualitative at best and quite arguable if one considers the OAM dispersion curves. The authors omit to mention and explain the discrepancies between simulated and experimental dispersion of OAM. The NMF extracted contributions of π^* show a rather asymmetric dispersion in Figure 2 but also in Figure S3, with a shoulder towards positive values of \hbar . This is also the case in Figure S2. For the σ^* hybrid, more intensity is towards positive \hbar in Figure 2, but towards negative \hbar in Figure S3. These deviations from a perfectly symmetric dispersion, expected from theory should be discussed further.

In the case of a simulated OAM-EELS experiment (Figure 2c), the σ^* component after NMF analysis perfectly overlaps with the true states calculated from DFT over the full energy range displayed. This is not the case for the π^* component, which

starts to deviate around the energy of the σ^* hybrid. How is this interpreted? Is this because both π^* and σ^* signals overlap above ~ 197 eV? Can this be interpreted as a sign that the σ^* component is only imperfectly extracted by NMF?

The residual component has the shape of the near edge structures, as underlined in the results section. This raises questions as to the meaning of the residuals. Could the authors indicate the relative intensity of each component (residuals, π^* , σ^*)? As the plots are shown, one would assume that all three components are reproduced without intensity scaling, please specify. If that is the case, the meaning of the high intensity residual containing significant spectral fine structures should be discussed. Specifically, how does a residual signal containing meaningful π^* and σ^* signatures affect the interpretation of the isolated π^* and σ^* NMF components?

- What is the reason for using a rather “thick” h-BN sample instead of a instead of few-layer sample as is typical for such archetype 2D material? In addition, the description of the OAM spectrum by the sorter is an incoherent sum, justified by the fact that the sample of interest is made of light elements. What additional limitations would appear for the interpretation of OAM-spectra from samples with heavier elements?

Reviewer #3

(Remarks to the Author)

In this manuscript, Tavabi and coworkers describe a novel and very interesting method of angular-momentum-resolved electron energy-loss spectroscopy. They demonstrate the use of an orbital angular momentum sorter in combination with electron energy-loss spectroscopy (EELS) to separate the contributions to the energy-loss spectrum linked to transitions with different orbital angular momentum (OAM). This is done by investigating the B K-edge of hexagonal boron-nitride as an example material.

In the well-written and clear manuscript the interpretation of the experimental results is backed by a comparison with simulations of the corresponding measurements. As the authors state, the perfect alignment of the electron microscope including the OAM sorter plates and the sample is necessary to acquire data with optimal quality. The difficulty of this procedure, combined with the restriction posed on the sample to be relatively thin and perfectly aligned while being stable enough to withstand the long acquisition times, might restrict the use of this technique to a certain class of materials. However, I think the described method is a valuable addition to other sophisticated EELS methods and provides great potential for investigations within different fields including materials science, where, e.g., information about magnetic properties are of interest. Thus I recommend the publication of the paper describing this novel approach to investigate OAM within EELS in Nature Communications.

To further improve the discussion, the authors should discuss the following question in their revision:

Why is the maximum of the experimental OAM profile components corresponding to OAM transitions of $l=\pm 1$ h located at an OAM of 3-4 h while some of the simulations show a maximum at $l=\pm 1$ h as expected from a theoretical point of view?

Version 1:

Reviewer comments:

Reviewer #1

(Remarks to the Author)

Comments on the re-submission of the manuscript entitled “First demonstration of angular momentum-resolved electron energy-loss spectroscopy”

The revised paper has improved the presentation of the experimental results by including the OAM-deconvoluted spectra in the main text and moving the statistical analysis to the supplementary materials. It however also reveals the main issues of the EELS using OAM sorters, shown clearly as a minor unexpected peak just above 190eV in Figure 4b, after the OAM broadening effects due to the finite instrumental resolution and the delocalization of the probe within the sample are supposed to have been taken into account. As a result, the applicability of EELS by OAM sorters as a quantitative spectroscopic tool merits more critical discussion than what has been presented. These and other technical issues need to be clarified before the revised manuscript can be published. Here are some specific comments:

1) For deconvolving the finite instrumental OAM resolution function, have the authors taken the chromatic aberration of the OAM sorter into account ?

2) The broadening of the OAM spectrum due to off-axis excitation has been well known for both optical and electron cases. The authors have cited reference 39 for the optical case and reference 40 for the electron beam case. The author should cite an early paper specifically addressing this delocalization issue in the OAM-resolved EELS (PHYSICAL REVIEW A 88, 031801(R), 2013).

3) In estimating the extent of the OAM spectral broadening due to off-axis excitation of a single atom, the authors have an interesting expression $\langle y \rangle = \sqrt{\sigma_p^2 + \sigma_x^2}$ from the uncertainty principle. The authors have taken the probe defocus as the only fitting parameter for σ_p , and the effective atom size (σ_x) is taken from the exit wave simulation. Can the authors comment on the variation of the beam wavefunction during its passage within the crystal

and its effect on the estimation of the effective size of the scattering atom (σ_x)? The formula is essentially a convolution of the broadened probe with a finite sized atom. In estimating the contribution of the neighboring atoms, is the asymmetric illumination of the off-axis atom also taken into account? Or put in another way, does the atom separation matter?

4) In the illustration for the Monte Carlo simulation, the inelastic scattering is assumed to take place in the middle of the crystal in the example of Fig. S3 and S4. Is this also true for the model-based decomposition approach, leading to the result shown in Fig. 4b? Some of such details need to be clarified and included in the Methods section of the main text.

5) The result of this estimation and that mentioned in the previous comment, is shown in Fig. 4b and judging from the pre-peak at near 190eV energy loss alone, would the authors agree that the agreement with Figure 4c is qualitative, rather than quantitative?

6) In Figure 4a, can the authors confirm that what are plotted are $c_m(\Delta E)$ mentioned in line 104? As $c_m(\Delta E)$ plays a central role, maybe it should be defined in a separate line of equations. Also in the discussion, the authors should comment on discrepancy and the cumbersome (model-based) process required to arrive at $c_m(\Delta E)$, particularly in estimating the delocalization effect due to elastic scattering of the probe within the crystal, and its suitability for unknown samples of defects within known crystal structures, or for magnetic dichroism application (mentioned in line 166) where the dichroic difference is typically minute..

7) Line 148-153 highlights the scanning OAM-EELS capability. I am not sure about the lesson to be learned from the example of scanning the sample edge where folding may be involved (according to the supplementary text). As folding involves tilting the OAM symmetry axis of the crystal away from the electron beam axis, complicating the interpretation. Without detailed structural models, such results can only be qualitative at best. This limitation should be acknowledged in the discussion.

Reviewer #2

(Remarks to the Author)

I thank the authors for their clear answers to my comments and those of the other reviewers, and for their thorough adjustments of the manuscript. I consider that all comments have been addressed appropriately and convincingly. Although the impact of the OAM sorter for EELS remains to be determined after future technical improvements, the experimental demonstration of OAM-EELS in this article is impressive and should be published.

As a minor comment, some experimental details of interest appear to be missing and would be of interest for the reader: an estimation of the experimental probe size for the first (5.4 mrad convergence semi-angle) and the second experiment (1.8 mrad convergence semi-angle on the probe-corrected instrument), the accelerating voltage for the first experiment, etc.

Reviewer #3

(Remarks to the Author)

The authors have revised the manuscript and added the application of model-based fitting to the experimental results. In my eyes, this improves the clarity of the manuscript. At this stage I do not have further comments and believe the manuscript can be published in its current form.

Version 2:

Reviewer comments:

Reviewer #1

(Remarks to the Author)

The revised version has addressed all the comments I have raised in my previous review and it is in a state fit for publication.

Modena, Jan. 3rd, 2025

Thank you very much for inviting us to revise our manuscript entitled "First demonstration of angular-momentum-resolved electron energy-loss spectroscopy". We have responded to the comments of the three reviewers by clarifying many details and improving the discussion. We have considerably strengthened the analysis of the data by using model-based fitting, which includes physics-based constraints. As suggested by Reviewer #1, we have used multivariate statistical analysis to provide further verification in the Supporting Information. We have added a discussion of the comparison with previous state-of-the-art EELS techniques, such as STEM-EELS with high convergence (or orbital mapping) and linear-momentum-dispersion-resolved EELS (qEELS). As suggested by Reviewer #2, we have discussed two strategies that could be used to improve the resolution by reducing higher OAM contributions to the spectrum. Our point-by-point response to the reviewers' comments is appended below. We have also followed the other editorial instructions.

We hope that the revised manuscript is suitable for publication in your journal.

Sincerely yours on behalf of all of the authors,

Giovanni Bertoni

Point by point replies to the reviewers' comments

Reviewer #1 (Remarks to the Author):

The paper presented experimental results on the orbital angular momentum resolved electron energy loss spectrum and their theoretical analysis. Despite the less than ideal instrumental resolution on the OAM state, the raw experimental data does show signs that the pre-edge spectral feature is associated with the dipole electronic transition involving exchange of zero unit of angular momentum and some of the post-edge spectral weight is associated with exchange of one unit of angular momentum. The results of multivariate statistical analysis merely confirm what one already can deduce qualitatively from the raw data, but adds additional confusion because it is well-known that the physical nature of different components are unclear in such statistical analysis. The discussion of the contribution to the OAM-resolved spectral signals covers possible OAM broadening through crystal diffraction and contribution from off-axis atoms, although I have some questions about the underlying assumptions, interpretation and implications. Overall, it is an interesting paper worthy publication after some essential modification. Below are my detailed comments:

1) Line 31: 'In the second quantization framework, the absorption of a virtual photon produced by a fast incident electron is approximately equivalent to real photon absorption' This reference to the second quantization framework is obscure, can the author provide a reference or more context to clarify the statement.

Our reply:

We have rephrased this sentence and added a proper reference, as follows:

"In quantum electrodynamics, the absorption of a virtual photon produced by a fast incident electron is approximately equivalent to real photon absorption⁵."

5. Lourenço-Martins, H., Lubk, A. & Kociak, M. Bridging nano-optics and condensed matter formalisms in a unified description of inelastic scattering of relativistic electron beams. *SciPost Physics* **10**, 031 (2021).

2) The raw experimental results are not reproduced in the main text, but Fig. 2 instead shows the spectral components deduced from non-negative matrix factorization algorithm processing (NMF). Given that NMF is a variation of the multivariable statistical analysis and I assume that it also suffers the usual problem of interpretability of the origins of the different decomposed components, it is not appropriate to show them as the main result in Fig. 2, instead of the raw data (shown in Fig. S1). As an example, can the authors explain the OAM dependence of the residual, which also display the same dependence as the so-called π^* components. In fact, what is the basis of identifying two similar looking components differently (the display of the scree plot in the supplementary materials could be useful here)? Incidentally, are the spectral intensities presented in Fig. 2a and 2c normalized in any way? If not, it is disturbing that the 'residual' has such a large intensity, indicating the significant failure of the OAM separation there.

3) Instead of NMF analysis in the main text (which is perfect for inclusion in SI), it maybe more profitable to present to present the raw data and a 'sharpened' version by deconvoluting out the instrumental OAM 'response function', given one is already collected at the zero energy loss (Fig S6b) and at pre-edge energy (S1b), or one can be recorded separately using the phase plates for different OAM channel. Alternatively, the raw OAM-EELS data can be compared directly with the properly convolved OAM-EELS spectrum from first principle calculation.

Our reply:

We reply to points 2) and 3) together. We agree that the extracted components from multivariate statistical analysis (such as NMF decomposition) can be difficult to interpret. The only real physical constraint is the positivity of the spectral features in the NMF decomposition. In the revised manuscript, we have moved the NMF decomposition to the Supporting Information, as suggested by the reviewer. We have also applied

model-based fitting to the data. This procedure allows the $m=0$ and $m=\pm 1$ components to be retrieved from the EEL spectrum. As suggested by the reviewer, the new version of Fig. 2 shows the raw experimental data after background subtraction alongside the data after deconvolution using the experimental psf from the zero-loss spectrum. The new version of Fig. 3 shows the results after final fitting of the data. The Results section on page 4 and the Discussion section on pages 4-7 have been rewritten. A paragraph has been added to the Methods section on page 8. The model details are described in section S3 in the Supporting Information, together with new Supporting Figs S3 and S4.

4) In lines 136-141, the authors discussed the extraction of bond orientation experiment qEELS experiments based on high momentum resolution and stated correctly that the use of nearly parallel illumination is incompatible with the use of atomic sized probes. In this context, the authors should also mention and comment on the convergent beam variation of the qEELS work, for example, that of ref 40 (an edge-on sample required), Hu et al. (2008) *Ultramicroscopy*, Vol 108, p405 (an edge-on sample not essential) and Xu et al (2023) *Phys. Rev. Lett.* 131, 186202 (an edge-on sample not required). It provides a complementary alternative to spatially resolved OAM-EELS analysis. This also applies to the relevant comments in lines 142-144 and 160-162.

Our reply:

We are grateful to the reviewer for this comment. We have extended the discussion with state-of-the-art EELS aiming at separating different transitions (or orbitals) in the spectrum on page 7. We distinguish between:

- a) STEM-EELS with partially-integrated q from the high convergence of the STEM probe, which relies on the large separation of features in the EEL spectrum to map the orbitals in space (so-called orbital mapping);
- b) qEELS, which acquires a linear-momentum-dispersed EEL spectrum and is used to reconstruct the angular dependence of the different orbitals in reciprocal space.

We have added the suggested reference for a plane-oriented sample in the first group:

48. Xu, M., Li, A., Pennycook, S. J., Gao, S.-P. & Zhou, W. Probing a defect-site-specific electronic orbital in graphene with single-atom sensitivity. *Phys Rev Lett* **131**, 186202 (2023).

We have added a further reference to qEELS:

54. Arenal, R., Kociak, M. & Zaluzec, N. J. High-angular-resolution electron energy loss spectroscopy of hexagonal boron nitride. *Appl Phys Lett* **90**, (2007).

We have also added the suggested reference:

43. Hu, X., Sun, Y. & Yuan, J. Multivariate statistical analysis of electron energy-loss spectroscopy in anisotropic materials. *Ultramicroscopy* **108**, 465–471 (2008)

to the discussion on page 6, where we refer to the experiment described in the Supporting Information. Even if it can be categorized in the STEM-EELS category above, this reference deals with principal component analysis and is a proper comparison to our analysis presented in Fig. S2.

5) In Figure 3, the consequence of inclusion of the inelastic scattered waves from the off-axis atoms is displayed qualitatively. Do you mean schematically? Given it is the single most important effect in canceling out the OAM-specific effect in a lot of EELS experiments tried so far, can we have a specific example, for example, from your Monte Carlo simulation mentioned in lines 98-110.

Our reply:

The new Fig. 4 shows quantitatively the results of the Monte Carlo method for 3 different atoms, one on-axis with respect to the probe and sorter axis and two slightly off-axis. The Monte Carlo method, which is based on a simplified version of a full multislice calculation, is described in section S3 of the Supporting Information. New Supporting Figs S3 and S4 are used to better explain the model.

6) Assume that the off-axis contribution is included in the simulated OAM spectra Fig. 2d, can we also include the theoretical OAM spectra from the on-axis atom alone for comparison? I think that the data can be readily extracted from Fig. 1c.

Our reply:

The simulated OAM spectra are presented in the new Fig. 3. The single atom is shown in the new Fig. 4.

7) A comparison of Fig. 1c and Fig. 2d seems to suggest that the observed broadening of the OAM distribution is largely due to the unavoidable contribution of the off-axis contribution of the neighboring atoms. Given that, how can the unknown non-zero OAM spectral modes be deduced correctly? (This is the prerequisite to realize the potentials stated in lines 147-156).

Our reply:

This has been made possible by using model-based fitting after simulation of the OAM profiles (see the new Fig. 3). The model is explained in the new Fig. 4 and in section S3 of the Supporting Information.

8) In Line 163, the prospect of multipolar transition is mentioned. Is there evidence for such transitions in the current B K-edge data? How can such a transition be distinguished from dipolar contribution from off-axis atoms?

Our reply:

The reviewer is correct. We have decided to limit the model fitting to dipolar contributions (i.e., $\Delta m = 0$ or $\Delta m = +1, -1$) because we did not find any evidence for multipolar features in our B spectra. It is possible to include higher multipolar transitions in the model (e.g., $\Delta m = +2, -2$ by using K_2 Bessel functions) to account for their contribution to the spectra. A lower variance in the OAM dispersion may be required. We have proposed two strategies in the Discussion section on page 6:

- a) Developing a new generation of sorter device that is optimized for high convergence;
- b) Developing a new generation of sorter device with an embedded aperture to limit the angular acceptance of the sorter.

9) The phrase 'diffraction' has been used in two different contexts: one of them is used properly as in Lines 111-112, '... no diffraction effect.', i.e. the diffraction by the rest of the crystal post inelastic core loss scattering. On the other hand, the term 'diffraction' has been used to indicate the Fourier transform of the exit wave by the objective lens (line 333). The second usage is confusing, hence please amend.

Our reply:

We have amended the text.

10) Given the discussion, can the authors summarize at the end of the paper the essential experimental conditions for the observation of intrinsic OAM-resolved EELS? For example, what is the maximum thickness before the crystal diffraction needs to be taken into account?

Our reply:

it is difficult to provide practical conditions for an OAM-resolved EELS experiment that are valid for a general case. Elastic scattering depends on the weight of the atoms and on the orientation and thickness of the sample. We have previously used simulations (see ref. 26 of the manuscript) to demonstrate that the effect of the crystal can be neglected up to a thickness of ~ 40 nm (for ferrite). We have clarified this point on page 4 of the Results section.

Can the OAM-EELS experiment have a simple interpretation if the symmetry axis of the sample does not align with the beam axis ?

Our reply:

A misalignment between the sample's symmetry axis and the beam axis introduces a lateral displacement of the inelastically-scattered exit wave with respect to the reference axis of the sorter. (A tilt in real space corresponds to a shift in Fourier space). Such a displacement directly impacts the definition of angular momentum, which is inherently linked to the choice of reference axis. As a result, such a misalignment leads to a broadening of the measured OAM spectrum, which in turn complicates the interpretation of the experimental results. (See, for instance, ref. 35 in the manuscript).

Reviewer #2 (Remarks to the Author):

In this manuscript, the authors report an experimental demonstration of an orbital-angular-momentum sorter for EELS spectra in the TEM. The authors chose to acquire data on a highly anisotropic material, hexagonal BN, which features π^* and σ^* contributions energies separated by several eV. The non-negative matrix factorization statistical analysis approach was chosen to exploit the experimental data. The manuscript is well written, contains a short results section followed by a longer discussion about the interpretation of the results, the current limitations and perspectives. Although this demonstration is a technical tour de force, which validates the predictions from simulations, I have several comments below.

- This is a rather compelling experimental demonstration of angular-momentum-resolved EELS on a ~ 8 nm thick h-BN sample. This manuscript provides ground to the OAM-EELS approach, which has been introduced on a theoretical basis by the authors in previous publications and conference abstracts. The demonstration that the OAM sorter is working in a TEM has already been reported by the same group of authors (reference 18, Amir H. Tavabi et al., physical review letters 126, 094802 2021), and since the OAM-EELS theory is also available in literature, the highlight of this manuscript is really on the experimental demonstration of an OAM-EELS on h-BN.

The added-value of the experiment is evident to validate the simulations. Nevertheless, beyond this experimental demonstration, the manuscript does not provide much support of the expected impact that the OAM approach could bring when probing localized states, decorrelating and mapping hybrid orbitals, as mentioned in the main text. This is rather disappointing since these ideas are introduced at the beginning (Figure 1), but remain to be shown experimentally.

One of the major downside of this experimental demonstration is the lack of spatial resolution, which is one of the main advantages of using a TEM. As such, the sub-nanometer or atomic-scale resolution available in current TEM instruments is the missing ingredient for a truly impactful demonstration. The discussion

contains the current limitations of the OAM-EELS approach and provides perspectives for this work. This is interesting, but at the same time clearly suggests that experimental difficulties remain to unleash the potential of this elegant approach, which is at the expense of the impact of the results. Beam scanning, for instance, is mentioned but not demonstrated because of physical or technical limitations of the sorter. The lack of clear broad impact perspective that one could expect for a journal of this caliber is detrimental. Whilst the separation of OAM components is of interest, the broad impact of the OAM sorter for TEM analysis is not apparent in this manuscript. As a result, and again despite the scientific soundness of the work, this manuscript would be a much better fit to the audience of a more specialized journal.

Our reply:

This is a very challenging experiment. We stress that our results demonstrate the first application of OAM and EELS double dispersion and confirm that the orbital angular momentum from inelastic transitions can be separated. In the revised manuscript, we show that we are close to the expected physical limits by comparing the standard deviation of the measured OAM to that obtained by modeling (reported in the Discussion session on page 6). Local probing of these OAM components opens possibilities for mapping magnetic or hybridized states at atomic resolution. However, it requires a technological improvement of the device for working at higher convergence angles. Further developments are beyond the scope of this paper. In the Discussion session on page 6, we have added two future strategies:

- a) Developing a new generation of sorter device that is optimized for high convergence,
- b) Developing a new generation of sorter with an embedded aperture to limit the angular acceptance.

Technological improvements in transmission electron microscopy have generally required time for full development and routine use. Previous examples include aberration correctors, energy monochromators and direct electron detectors, which each required at least a decade from first demonstration to routine use.

- Overall, the results in Figure 2 should be explained and discussed further in the manuscript, as detailed in the following:

The use of the word "closely" in the sentence "The NMF-extracted experimental components closely mirror those predicted by DFT calculations..." is qualitative at best and quite arguable if one considers the OAM dispersion curves. The authors omit to mention and explain the discrepancies between simulated and experimental dispersion of OAM. The NMF extracted contributions of π^* show a rather asymmetric dispersion in Figure 2 but also in Figure S3, with a shoulder towards positive values of \hbar . This is also the case in Figure S2. For the σ^* hybrid, more intensity is towards positive \hbar in Figure 2, but towards negative \hbar

In Figure S3. These deviations from a perfectly symmetric dispersion, expected from theory should be discussed further.

In the case of a simulated OAM-EELS experiment (Figure 2c), the σ^* component after NMF analysis perfectly overlaps with the true states calculated from DFT over the full energy range displayed. This is not the case for the π^* component, which starts to deviate around the energy of the σ^* hybrid. How is this interpreted? Is this because both π^* and σ^* signals overlap above ~ 197 eV? Can this be interpreted as a sign that the σ^* component is only imperfectly extracted by NMF?

The residual component has the shape of the near edge structures, as underlined in the results section. This raises questions as to the meaning of the residuals. Could the authors indicate the relative intensity of each component (residuals, π^* , σ^*)? As the plots are shown, one would assume that all three components are reproduced without intensity scaling, please specify. If that is the case, the meaning of the high intensity residual containing significant spectral fine structures should be discussed. Specifically, how does a residual signal containing meaningful π^* and σ^* signatures affect the interpretation of the isolated π^* and σ^* NMF components?

Our reply:

As also noted by Reviewer #1, the components that are extracted using multivariate statistical analysis often have no clear physical meaning and are difficult to interpret. These issues are highlighted by the discrepancies between the simulated profiles and the π^* extracted profile. For this reason, the new Figs 2 and 3 apply model-based fitting to the experiment results. We have also rewritten the Results session on page 4 and the Discussion session on pages 4-7. Furthermore, the details of NMF decomposition, which have been moved to the Supplementary Information, provide further proofs that OAM decomposition can be used when the physical details are not known (e.g., the experimental point spread function, probe function and sample thickness).

- What is the reason for using a rather “thick” h-BN sample instead of a instead of few-layer sample as is typical for such archetype 2D material? In addition, the description of the OAM spectrum by the sorter is an incoherent sum, justified by the fact that the sample of interest is made of light elements. What additional limitations would appear for the interpretation of OAM-spectra from samples with heavier elements?

Our reply:

We are interested in spectral features that originate from hybridization of the constituent atoms but are independent of the thickness of the sample. The primary reason for using a bulk h-BN sample of thickness

~8 nm is to ensure a sufficient signal in the B K-edge by increasing the probability of single inelastic scattering events, while keeping multiple scattering negligible. The incoherent nature of the sum of OAM components from different B atoms in our simulations results from the inelastic scattering events being incoherent and does not depend on atomic weight. For a larger specimen thickness or heavier elements, dynamical effects will introduce features in the OAM spectrum that are related to the symmetry of the crystal. A detailed description of the method used for simulating OAM profiles is included in section 3 of the Supplementary Information.

Reviewer #3 (Remarks to the Author):

In this manuscript, Tavabi and coworkers describe a novel and very interesting method of angular-momentum-resolved electron energy-loss spectroscopy. They demonstrate the use of an orbital angular momentum sorter in combination with electron energy-loss spectroscopy (EELS) to separate the contributions to the energy-loss spectrum linked to transitions with different orbital angular momentum (OAM). This is done by investigating the B K-edge of hexagonal boron-nitride as an example material.

In the well-written and clear manuscript the interpretation of the experimental results is backed by a comparison with simulations of the corresponding measurements. As the authors state, the perfect alignment of the electron microscope including the OAM sorter plates and the sample is necessary to acquire data with optimal quality. The difficulty of this procedure, combined with the restriction posed on the sample to be relatively thin and perfectly aligned while being stable enough to withstand the long acquisition times, might restrict the use of this technique to a certain class of materials. However, I think the described method is a valuable addition to other sophisticated EELS methods and provides great potential for investigations within different fields including materials science, where, e.g., information about magnetic properties are of interest. Thus I recommend the publication of the paper describing this novel approach to investigate OAM within EELS in Nature Communications.

To further improve the discussion, the authors should discuss the following question in their revision:

Why is the maximum of the experimental OAM profile components corresponding to OAM transitions of $l=\pm 1$ h located at an OAM of 3-4 h while some of the simulations show a maximum at $l=\pm 1$ h as expected from a theoretical point of view?

Our reply:

We are grateful to the reviewer for the positive comments about our manuscript. As noted also by reviewers #1 and #2, multivariate statistical analysis extracts components that often have no clear physical interpretation. This issue is evident in the discrepancies in our extracted experimental profiles. It is a limitation of multivariate statistical analysis approaches (such as NMF) and is visible also in components

extracted from simulations. For this reason, we have rewritten the Results section and the Discussion section and changed the figures in the revised manuscript based on the application of model-based fitting to our experimental results.

Point by point replies to the reviewers' comments

Reviewer #1 (Remarks to the Author):

Comments on the re-submission of the manuscript entitled "First demonstration of angular momentum-resolved electron energy-loss spectroscopy"

The revised paper has improved the presentation of the experimental results by including the OAM-deconvoluted spectra in the main text and moving the statistical analysis to the supplementary materials. It however also reveals the main issues of the EELS using OAM sorters, shown clearly as a minor unexpected peak just above 190eV in Figure 4b, after the OAM broadening effects due to the finite instrumental resolution and the delocalization of the probe within the sample are supposed to have been taken into account. As a result, the applicability of EELS by OAM sorters as a quantitative spectroscopic tool merits more critical discussion than what has been presented. These and other technical issues need to be clarified before the revised manuscript can be published. Here are some specific comments: [...]

We thank the reviewer for considering we improved the presentation of the results. We first comment on the general applicability of the OAM-EELS technique.

In summary, our technique is fundamentally robust, however, due to the inherent nature of inelastic scattering, a model-based fitting approach is necessary for accurate interpretation of the results.

In this work, we present a simplified model by assuming a continuum atomic distribution. This approximation is justified in our specific experiment by the presence of sample drift and a large probe, which is a technical limitation of the present setup rather than an intrinsic limitation of the method. Importantly, for a thin sample where dynamical diffraction effects remain negligible, the model is valid even in the case of a steady sample or, conversely, in the case scanning mode (STEM) is used to aim the beam at given atomic positions. In such conditions, the Monte Carlo approach used here can be replaced by a discrete sum over atomic positions without altering the core principles of the analysis presented here.

When dynamical effects become significant, such as in thicker samples where multiple scattering plays a dominant role, our simplified model is no longer sufficient. In these cases, a more rigorous multislice simulation would be required. However, such an approach is already well established in literature. Thus, while the complexity of the modelling increases in these scenarios, the fundamental considerations regarding OAM-resolved EELS remain unchanged.

Ultimately, the OAM-EELS technique itself does not impose any intrinsic limitations on applicability. The primary challenges stem from the physics of inelastic scattering, which affect any EELS technique. Compared to alternative methods, OAM-EELS offers a more complete measurement of angular momentum transfer, capturing not only ± 1 components (as for instance in EMCD) but also higher-order contributions, making it a powerful tool for future investigations of inelastic electron

scattering at the atomic scale. The OAM sorter is indeed, in a quantum metrology sense, the optimal measurement of OAM.

The applicability has been more critically discussed throughout this revision, and it is detailed in the point-by-point discussion below.

1) For deconvolving the finite instrumental OAM resolution function, have the authors taken the chromatic aberration of the OAM sorter into account?

We thank the referee for this question, which provides an opportunity to clarify a particularly delicate aspect of our experimental setup. In our study of the Sorter, we have extensively examined the impact of various aberrations on OAM resolution, including chromatic aberration. All these effects are included in our simulation software, as described in a previous paper (ref. 22), and the code is freely available at <https://doi.org/10.5281/zenodo.4770744>.

Here, we present the influence of chromatic aberration on the OAM Sorter's 2D psf (top row) and 1D psf (bottom row). We recall that the energy spectrometer's optics integrate the beam along the direction orthogonal to the OAM (i.e., the radial coordinate) while dispersing it in energy.

The calculations were performed at different energies, with zero-loss serving as a reference. For the B K-edge (190 eV) used in this study, we observe that the effect of chromatic aberration is mainly along the radial direction and therefore has negligible influence on OAM resolution. This suggests that we have at least a 400 eV energy range, centred on the energy at which the sorter is aligned, free from significant chromatic aberration. Given the typical energy dispersion in EELS experiments, this window is sufficiently large for most applications. To observe a noticeable chromatic effect, an energy loss as high as 400 eV must be considered. Even at such large energy loss, it is worth noting that the impact of chromatic aberration is comparable to a simple defocus (df/f), as previously

described in the supporting information of ref 22. Consequently, it can be largely compensated for by a small defocus adjustment (e.g., $df/f = 0.2\%$ of the objective lens focal distance f at 400 eV). In conclusion, chromatic aberration is not a significant factor, and we have neglected its effect in the present application. It becomes relevant only if the beam is not well centred on the objective lens, as this would introduce a chromatic shift that could potentially disrupt the alignment of the sorter. However, beam centring is an integral part of the standard microscope alignment procedure (rotation centre alignment).

To clarify this, we have added the following line in the manuscript at p. 3:

“[...] and since chromatic effects of the sorter can be neglected in this energy range (see Supplementary Information), [...]”.

Additionally, we have included a section in the Supplementary Information detailing the chromatic effects along with the figure shown above.

2) The broadening of the OAM spectrum due to off-axis excitation has been well known for both optical and electron cases. The authors have cited reference 39 for the optical case and reference 40 for the electron beam case. The author should cite an early paper specifically addressing this delocalization issue in the OAM-resolved EELS (PHYSICAL REVIEW A 88, 031801(R), 2013).

We thank the reviewer for their advice. We have added the suggested reference as new ref. 40 and renumbered subsequent references in the revised manuscript:

“40. Yuan, J., Lloyd, S. M. & Babiker, M. Chiral-specific electron-vortex-beam spectroscopy. *Phys Rev A (Coll Park)* **88**, 031801 (2013).”

3) In estimating the extent of the OAM spectral broadening due to off-axis excitation of a single atom, the authors have an interesting expression $\langle y \rangle = \sqrt{\sigma_p^2 + \sigma_x^2}$ from the uncertainty principle. The authors have taken the probe defocus as the only fitting parameter for σ_p , and the effective atom size (σ_x) is taken from the exit wave simulation. Can the authors comment on the variation of the beam wavefunction during its passage within the crystal and its effect on the estimation of the effective size of the scattering atom (σ_x)? The formula is essentially a convolution of the broadened probe with a finite sized atom. In estimating the contribution of the neighboring atoms, is the asymmetric illumination of the off-axis atom also taken into account? Or put in another way, does the atom separation matter?

The formula for the variance of the expected OAM profile (σ_ℓ) in eq. 3 at p.6 provides a broad estimation based on a continuum approximation of the crystal, by assuming that scattering can occur at any location within the probed area. The main goal of this expression is to account for off-axis

atomic positions, which are captured through the term $\langle y \rangle$ spanning the probe size. In this regard the in-plane separation of the atoms does not matter. The exit wave primarily determines the value of σ_x , reflecting the spatial extent of the scattered wavefunction at the exit surface of the sample and, as such, it is affected by the in-depth atom separation.

This formulation highlights the trade-off between off-axis scattering and the spatial extent of the exit wave: if the exit wave is broad, off-axis dispersion plays a reduced role in the overall OAM broadening. However, the formula itself does not explicitly include effects related to atomic coordination and preferred interatomic distances in a crystal, nor does it account for the evolution of the probe wavefunction as it propagates through the material. The probe evolution is entirely accounted for in the measurement of σ_x through multislice simulations. However, we note that it can be generalised with the 1s approximation by considering the scaling of the 1s Bloch states as $\sigma_x \propto \sqrt{\frac{Z}{d_z^{5/4}}}$, where Z is the atomic number, and d_z is the interatomic distance in the atomic column (see ref. 44 in the revision). In the figure below we plot the size of the scattered atom and the corresponding variance in the OAM profile calculated from eq. 3, as a function of the atomic number Z , considering the very same illumination of 5.4 mrad as in the present experiment. We have indicated two points for boron and iron as references.

The Figure shows that the expected variance in OAM decreases as a function of the atomic number, making it suitable for measuring other materials, as far as the 1s Bloch wave approximations holds, i.e., the sample is thin (e.g., few tens of nm for Fe). Moreover, the variance reaches 1 \hbar at $Z \geq 30$. In conclusion, while our approximation is not exact, it provides a good first-order description of the interplay between probe size, scattering position, and exit wave broadening.

We have added in the Discussion at p. 6 the following sentence:

“The formula in eq. 2 can be generalized for atomic number Z in the 1s Bloch wave approximation⁴⁴, where the focusing of the propagating beam due to the atomic columns can be semi-empirically

expressed as $\sigma_x \propto \sqrt{\frac{z}{d_z^{5/4}}}$, with d_z the separation of the atom in the column. Heavier atoms are expected to produce more peaked OAM profiles.”

We have also added the following reference:

“44. Van Dyck, D. & Op de Beeck, M. A simple intuitive theory for electron diffraction. *Ultramicroscopy* **64**, 99–107 (1996).”

4) In the illustration for the Monte Carlo simulation, the inelastic scattering is assumed to take place in the middle of the crystal in the example of Fig. S3 and S4. Is this also true for the model-based decomposition approach, leading to the result shown in Fig. 4b? Some of such details need to be clarified and included in the Methods section of the main text.

The model-based method leading to the results in Fig. 4 is based on a kinematic approach that exploits the 1s Bloch state approximation (also known as the linear approximation), a well-established method in TEM.

In this approximation, the interaction between the electron probe and the periodic crystal potential is simplified by considering only the dominant Bloch state of the crystal, specifically, the 1s Bloch state which represents a strong channelling mode that localizes the electron density along the atomic columns. This approximation is particularly valid for thin, light samples, such as the BN film investigated in this study.

Bloch states are inherently 2D functions, meaning they do not evolve along the atomic column. Instead, dynamic effects arise from the interference of multiple Bloch states. Within the kinematic regime of the 1s approximation, the concept of “depth” becomes meaningless, and the exit wavefunction of the electron can be described as a simple product of 2D functions, as illustrated in Figure S4.

We acknowledge the referee’s point that the detailed explanation of this model, provided in the Supplementary Information, was not adequately reflected in the Methods section.

To address this, we have added the following sentence in Methods section at p. 9:

“This approach relies on a kinematic model of the exit wave at the sample plane, where the exit wave is expressed as a simple product of 2D functions: the 1s Bloch state, the probe, and the atomic scattering factor, [...]”

5) The result of this estimation and that mentioned in the previous comment, is shown in Fig. 4b and judging from the pre-peak at near 190 eV energy loss alone, would the authors agree that the agreement with Figure 4c is qualitative, rather than quantitative?

The technique *per se* is quantitative, by selecting the final state according to its OAM. Even with the present sorter prototype, the agreement in Figure 4b,c should be considered at least semi-quantitative. The reviewer is probably referring to the small residual peak at 190 eV in the $m = \pm 1$ (σ^*) spectrum. Despite this peak, the overall intensities in the features agree well between the experiment and the theoretical predictions. To quantify the level of agreement, we report some parameters:

1) The ratio between the π^* peak maximum and the σ^* peak maximum is 1.73 for the experiment in Fig. 4b and 1.79 the theoretical calculation in Fig. 4c, with a discrepancy of 4%.

2) A metric for the similarity between two signals is given by the normalized cross-correlation (NCC). The NCC between two vectors \mathbf{a} and \mathbf{b} (the spectra to compare in our case) is defined as:

$$\text{NCC}(\mathbf{a}, \mathbf{b}) = \frac{\mathbf{a} \cdot \mathbf{b}}{ab} = \frac{\sum_i a_i b_i}{\sqrt{\sum_i a_i^2} \sqrt{\sum_i b_i^2}}, \quad (12)$$

and gives a quantitative measurement of the similarity between the two vectors. Ideal match is +1 when $\mathbf{a} = \mathbf{b}$, orthogonality is 0, and complete anticorrelation $\mathbf{a} = -\mathbf{b}$ is -1. The NCC between experimental and theoretical π^* and σ^* components (full lines in Fig. 4b) in the interval 185 – 220 eV, is 0.93 for π^* and 0.88 for σ^* , respectively. We excluded higher energies due to deviations in the *ab-initio* DFT calculations for high levels empty states. These values indicate high correlation between the experimental and theoretical spectra and are remarkable for a proof-of-concept demonstration obtained with a custom-built experimental setup.

3) If we assume that the peak at $m = \pm 1$ at 190 eV is due to a cross-talk between the $m = 0$ and $m = \pm 1$ channels, the experimental intensity at $m = \pm 1$ can be written as $I_{exp}(m = \pm 1) = c I_{theo}(m = 0)$, while $I_{exp}(m = 0) = (1 - 2c) I_{theo}(m = 0)$. The crosstalk coefficient in the deconvoluted spectrum results:

$$c = \frac{I_{exp}(m=1)}{I_{exp}(m=0) + 2I_{exp}(m=1)}. \quad (13)$$

In the spectra of Fig. 4b, $c = 11\%$. This incertitude is acceptable for many applications. For example, if we assume conservatively that the cross talk between +1 and -1 is $\leq 0.5c = 6\%$, which is sufficient to estimate magnetic dichroism values that may reach $\geq 18\%$ (see ref. 26).

Finally, if the source of discrepancy is a residual channelling effect, not considered by the simplified (kinematic) model of propagation used our model-based procedure, more rigorous modelling based on a full multislice could be explored in the future.

To address this, we have added the following sentence in the Results session at p. 5:

“The overall agreement between experimental and theoretical spectra demonstrates the quantitative nature of the method, with key spectral features matching both in relative intensity and shape. A small residual peak at 190 eV in the $m=\pm 1$ (σ^*) spectrum is observed, but its influence is minimal, as confirmed by quantitative metrics such as peak ratio analysis and normalized cross-correlation, as detailed in Supplementary Information. We estimate a crosstalk contribution of approximately

11%, which remains within acceptable limits for most applications, including magnetic dichroism measurements. Possible sources of this discrepancy include residual channelling effects, minor misalignments affecting the sorter psf, or limitations in the physical model. While our simplified model is sufficient for the present study, future refinements could incorporate a full inelastic multislice approach for enhanced accuracy.”

Additionally, we have included a section in the Supplementary Information detailing the quantitative analysis.

6) In Figure 4a, can the authors confirm that what are plotted are $c_m(\Delta E)$ mentioned in line 104? As $c_m(\Delta E)$ plays a central role, maybe it should be defined in a separate line of equations. Also in the discussion, the authors should comment on discrepancy and the cumbersome (model-based) process required to arrive at $c_m(\Delta E)$, particularly in estimating the delocalization effect due to elastic scattering of the probe within the crystal, and its suitability for unknown samples of defects within known crystal structures, or for magnetic dichroism application (mentioned in line 166) where the dichroic difference is typically minute.

We thank the referee for pointing this out. We clarify these aspects in the revisions, defining more rigorously the functions used in the formulas. To improve clarity, we have now explicitly defined $c_m(E)$ in a separate equation in the main text (eq. 2 at p. 4) and referred to them in the Figures:

“The simulated OAM profiles $\Gamma_m(\ell)$ shown in Fig. 3b were finally fitted to the experimental data $I(E, \ell)$, in order to obtain a set of coefficients $c_m(E)$ corresponding to EEL spectra at different m , according to the equation:

$$I(E, \ell) = \sum_m c_m(E) \Gamma_m(\ell). \quad (2)$$

Figure 4a shows the resulting reconstructed OAM-EEL spectrum $I(E, m) = \sum_m c_m(E)$ for comparison with Fig. 1c.”

As for the observed discrepancies between the experimental and theoretical spectra, these are now acknowledged and discussed in the main text (see reply to point 5). In particular, we provide a quantitative analysis of the agreement using cross-correlation and peak intensity ratios, demonstrating that the method is at least semi-quantitative despite residual differences.

In the last comment, the referee raises concerns about the necessity of a model-based approach, suggesting that this makes the method cumbersome. However, it is important to emphasize that model-based fitting is a fundamental requirement for *any* quantitative EELS analysis. Delocalization is indeed an intrinsic property of inelastic scattering and is not a specific artifact of our OAM-EELS

setup: inelastic scattering cross-sections are large, typically exceeding interatomic distances in a crystalline material.

The figure below compares the atomic inelastic scattering amplitudes in the case of B-K edge (~200 eV) and Fe-L edge (~700 eV) for the corresponding isolated atoms. For clarity we normalized the amplitudes with respect to their maximum:

As a result, interactions are never confined to a single atomic site but encompass multiple neighbouring atoms.

Indeed, technological limitations in our current experimental setup partly exacerbate the delocalization effect, but this issue can be mitigated through straightforward improvements in the fabrication of the first sorting element, which is simply a needle. Enhancing the precision of this component will enable the use of a larger beam-forming aperture, leading to a reduction in probe size while simultaneously increasing the beam current. This, in turn, will allow for shorter acquisition times, thereby minimizing sample drift and improving the overall stability and accuracy of the measurements. Moreover, the possibility of scanning the beam would enable localizing the probe on the atomic columns.

We can simulate this condition using the model, as reported in the image below. Using a 7 mrad probe (which is the standard value for uncorrected microscope at 300 keV) and locating the probe directly on top of an atomic column is sufficient to drastically reduce the delocalisation effect and obtain a clean spectrum where only the $m = 1, 0$ and $+1$ components are present. Further reducing the probe size using a 15 mrad aperture (which is instead a common value for Cs corrected microscopes) further improve the spectrum quality.

In these conditions, the interpretation of OAM spectra is more direct, due to a narrower OAM spread, but a degree of mixing between components is unavoidable and model-based deconvolution remain essential.

The approach we employ is well-established in literature and widely used in EMCD and other EELS-based techniques (see Ref. 33). The methodology accounts for elastic scattering by simulating the electron propagation through the sample before and after inelastic events, typically using a multislice or Bloch wave approach.

The suggestion that elastic scattering effects would make the method unsuitable for unknown samples is somewhat misplaced. It is unrealistic to perform an advanced experiment such as this without prior structural knowledge of the sample. Even conventional EELS requires structural context for interpretation. Moreover, the interatomic distances, which play the dominant role, are well-known and routinely measured in TEM experiments.

Future developments of OAM-EELS aim at scanning the probe to acquire *orbital maps*, enabling the study of defects and spatial variations in OAM signals. This will provide a more comprehensive understanding of local electronic structure.

Finally, for magnetic dichroism applications (e.g., EMCD), the key advantage of OAM-EELS is its ability to isolate specific final states, enhancing the sensitivity of dichroic measurements. Unlike conventional EMCD, where the dichroic signal is weak, OAM-EELS allows for near-complete isolation of the dichroic component. This leads to a significantly enhanced signal-to-noise ratio, making the detection of magnetic effects far more robust and reliable compared to other methods. [26,27]

We have clarified the effect of inelastic scattering at the beginning of the Discussion section at p.5: "At energies in the range of a few hundred eV, the inelastic scattering function has long-range tails that extend beyond the nearest-neighbour atomic distances. This means that atoms away from the

immediate probe position contribute to the scattering signal, reducing the localization of the interaction. Secondly, [...]"

And further at p. 6-7:

"The primary limitation of the current OAM-EELS experiment stems from the fabrication precision of the first sorting element. However, these are not fundamental limitations but rather technological challenges that can be overcome by improving the OAM sorter design so that it works with larger convergence semi-angles, thereby reducing the size of the probe and the value of σ_p . Similarly, limiting the collection angle by using an aperture built into the device, which is equivalent to increasing the uncertainty σ_x also has the effect of reducing the OAM broadening σ_L . While the intrinsic delocalization of inelastic scattering will always require a model-based fitting for rigorous quantitative interpretation (see Supplementary Information), a more advanced experimental setup will reduce artifacts and simplify this process, further strengthening the method's reliability and applicability."

7) Line 148-153 highlights the scanning OAM-EELS capability. I am not sure about the lesson to be learned from the example of scanning the sample edge where folding may be involved (according to the supplementary text). As folding involves tilting the OAM symmetry axis of the crystal away from the electron beam axis, complicating the interpretation. Without detailed structural models, such results can only be qualitative at best. This limitation should be acknowledged in the discussion.

We agree with the referee that interpreting the signal measured on the edge of the sample requires detailed structural models. Indeed, we did not attempt such analysis as it is well beyond the scope of this paper. The spatial mapping, which is only reported in the supplemental materials, serves the purpose of demonstrating the ability to achieve spatially resolved information with our technique.

It is important to highlight that scanning with the OAM sorter is not straightforward. Unlike conventional STEM-EELS, shifting the beam before the sample alters the alignment of the sorter, which must remain precisely calibrated. To compensate for this, we utilized the descans coils, as described in the Methods section, to maintain the correct alignment. While the present demonstration is limited to a simple case, such as the sample edge, we believe that establishing the feasibility of spatially resolved OAM-EELS measurements is a crucial step forward. Given that future advancements in the technique will require the ability to scan the beam for atomic-resolution experiments, this demonstration provides an essential proof-of-concept.

We have clarified the purpose of the scanning experiment in the Discussion section at p. 6:

"Achieving spatially resolved OAM-EELS measurements presents additional challenges compared to conventional EELS, as scanning the beam can disrupt the precise alignment of the TEM column and the OAM sorter. In particular, accurate alignment of the S1 and S2 phase plates is crucial to

maintain optimal performance, especially when scanning is required, such as during spectrum image acquisition. [...]

Interpreting in a quantitative way these observations require detailed structural models of the sample folding, and it is well beyond the scope of the present paper. However, the ability to measure a spatially resolved signal hints at future applications of atomic-resolution scanning experiments.”

Reviewer #2 (Remarks to the Author):

I thank the authors for their clear answers to my comments and those of the other reviewers, and for their thorough adjustments of the manuscript. I consider that all comments have been addressed appropriately and convincingly. Although the impact of the OAM sorter for EELS remains to be determined after future technical improvements, the experimental demonstration of OAM-EELS in this article is impressive and should be published.

As a minor comment, some experimental details of interest appear to be missing and would be of interest for the reader: an estimation of the experimental probe size for the first (5.4 mrad convergence semi-angle) and the second experiment (1.8 mrad convergence semi-angle on the probe-corrected instrument), the accelerating voltage for the first experiment, etc.

We thank the reviewer for their positive comment. We have already commented on the applicability of the sorter and clarify our plans for development in the replies to reviewer#1. We have included the missing experimental details (accelerating voltage and nominal probe-size) in the Methods section at pp. 8,9 as suggested.

Reviewer #3 (Remarks to the Author):

The authors have revised the manuscript and added the application of model-based fitting to the experimental results. In my eyes, this improves the clarity of the manuscript. At this stage I do not have further comments and believe the manuscript can be published in its current form.

We thank the reviewer for their positive comment.